# Building an AI-Driven Research Knowledge Graph for Problem Discovery and Organization

## Abstract

The rapid growth of scientific literature makes it increasingly difficult for researchers to identify open problems and track evolving opportunities. This paper proposes a vision for an AI-driven system that ingests research papers and transforms their content into structured, machine-navigable representations of open problems. By representing problem statements, assumptions, datasets, and constraints in a graph with semantic and citation-based relations, the system would enable novel queries and ranking mechanisms to surface high-value research opportunities. Importantly, this work is presented as a proposal and conceptual framework for future development rather than a description of a completed implementation. The contributions of this proposal lie in outlining the motivation, potential methodology, and expected impact of building such a research knowledge graph to support discovery, education, and collaboration across domains.

**Keywords:** AI-driven research discovery, research knowledge graph, large language models, research problem extraction, semantic retrieval, citation-based relations, automated hypothesis generation, ranking mechanisms, autonomous research agents, human-in-the-loop validation, reproducibility, mechanistic interpretability

## 1 Introduction / Background and Significance

The volume of scientific research published each year is growing at an unprecedented rate, making it increasingly difficult for researchers to identify open problems, track relationships between findings, and assess opportunities for contribution. While digital libraries and large language models (LLMs) have improved access to information, they are not optimized for systematically extracting structured research challenges or linking them across domains in a way that is actionable for researchers.

From the perspective of early-career researchers and PhD students, this challenge is especially acute. Extending existing research often requires reading hundreds of papers to uncover unresolved problems, identify assumptions and constraints, and map dependencies across works. For students who may be slower readers or who are new to a domain, this process is not only time-consuming but also a barrier to meaningful contribution. In effect, the pace of publication outstrips the ability of individual researchers to keep up, creating a growing gap between available knowledge and actionable research opportunities.

For example, a student investigating cache eviction policies may need to read dozens of papers across multiple venues just to discover that only a small fraction explicitly mention write amplification as an unresolved issue. Even once identified, those problems are scattered across sections such as "Discussion," "Limitations," or "Future Work," requiring substantial effort to synthesize. This inefficiency highlights the need for tools that can surface problems directly rather than forcing researchers to reconstruct them piecemeal from raw text.

This project addresses that gap by proposing the design and evaluation of an AI-driven system that ingests research papers and transforms their content into structured, machine-navigable representations of open problems. By representing problem statements, assumptions, constraints, datasets, and metrics as nodes in a graph, and connecting them through semantic and citation-based relations, we create a living repository of research opportunities.

The uniqueness of this approach lies in its focus on *problems* rather than *papers*. Rather than treating papers as the atomic unit of research, the system extracts the underlying challenges they describe, organizes them into a knowledge graph, and enables structured queries across domains. This shift makes it possible to ask questions that would be prohibitively time-intensive using current tools, such as:

- "*Show open problems in automated program repair that extend existing techniques and cite datasets with real-world bug reports.*"
- "*Rank research challenges in software testing by tractability and availability of benchmark suites.*"

It is important to emphasize that this work is currently a proposal and vision for future development, not a system that has already been implemented. The ideas outlined here represent a conceptual framework and research agenda that will guide subsequent design, prototyping, and evaluation.

In doing so, the system lowers the entry barrier for new researchers, accelerates literature discovery, and provides a foundation for sustaining the research community over time. By continuously surfacing open problems, linking them to relevant evidence, and enabling agents (or researchers) to replicate and extend experiments, the system supports an iterative cycle of discovery. This ensures that the research community remains dynamic, inclusive, and able to build cumulatively on prior work rather than losing opportunities in the flood of new publications.

## 2 Literature Review: AI-Driven Knowledge Graphs and Research Discovery

### 2.1 Scholarly Knowledge Graphs (SKGs) in Academia

Early work on scholarly knowledge graphs demonstrated the promise of structuring research outputs into machine-navigable formats. The **Open Research Knowledge Graph (ORKG)** Jaradeh et al. [2019] pioneered the idea of crowdsourcing structured representations of contributions such as research questions, methods, and results. While ORKG enables semantic comparisons across papers and aligns with FAIR data principles, its reliance on manual curation limits scalability. Later efforts shifted toward automation. The **Artificial Intelligence Knowledge Graph (AI-KG)** Dessì et al. [2020] mined over 330K papers to produce 820K entities describing AI research concepts, tasks, and results. This large-scale, automatically generated KG illustrated feasibility but also revealed challenges in accuracy, particularly with entity linking. Building on this, the **Computer Science Knowledge Graph (CS-KG)** expanded coverage from AI to all of computer science, scaling from 6.7M papers in its initial release to 15M in CS-KG 2.0 Dessì et al. [2022], Meloni et al. [2025]. These resources support trend analysis, hypothesis generation, and semantic search, but they remain focused on papers and claims rather than the explicit extraction of open problems. Domain-specific KGs such as *SoftwareKG* Schindler et al. [2020] and broad bibliographic indices like *OpenAlex* Priem et al. [2022] highlight the diversity of approaches. However, aligning metadata-focused graphs with content-focused extractions remains a challenge. The novelty of our proposal lies in its emphasis on structuring *research problems* as first-class objects, moving beyond claims and bibliographic metadata toward actionable research opportunities.

### 2.2 Extraction of Problems and Research Gaps from Text

Parallel to SKGs, a growing body of work has focused on extracting research questions and gaps directly from text. Taslimi et al. (2025) Taslimi et al. [2025] present a hybrid pipeline that combines heuristics, classifiers, and LLMs to detect explicit and implicit research questions. Resources such as *SciREX* Jain et al. [2020] enable document-level annotation of tasks, materials, metrics, and relations, demonstrating the importance of capturing context beyond individual sentences. Prior to LLMs, heuristic methods targeted high-yield sections like "Future Work" or "Conclusions," but these approaches achieved high precision at the expense of recall. More recently, LLMs have been leveraged for open problem extraction, sometimes combined with retrieval-augmented generation to suggest

future directions Gan et al. [2024]. A particularly relevant contribution is *HypoGen* Qi et al. [2025], which mined thousands of problem–hypothesis pairs from computer science papers, illustrating how generative models can capture idea evolution. While these studies highlight the feasibility of extraction, they stop short of representing problems in a graph with semantic and relational context. Our work is novel in that it unifies extraction with structured graph-based organization, allowing problems in software engineering to be queried, linked, and ranked across subfields such as testing, program repair, and requirements engineering.

## 2.3 Semantic Retrieval and Knowledge Discovery in Science

Once extracted, research knowledge must be made discoverable. Embedding-based retrieval models such as SPECTER capture conceptual similarity beyond surface-level keywords, improving classification and recommendation tasks. Hybrid retrieval approaches that combine vector similarity with lexical or metadata filters balance semantic breadth with precision. Graph-based retrieval has also emerged, leveraging heterogeneous entities such as authors, datasets, and methods to discover new links. For example, *ResearchLink* Borrego et al. [2025] integrates graph path features with embeddings to recommend hypotheses. At scale, projects like ORKG and CS-KG expose SPARQL endpoints, enabling structured queries over millions of triples. However, current retrieval systems remain oriented around papers and claims. By contrast, the novelty of our proposal is its direct support for problem-oriented queries—for example, asking specifically about unresolved challenges in software engineering tied to datasets, metrics, or constraints—thus offering functionality not captured by existing embedding- or graph-based retrieval approaches.

## 2.4 AI-Assisted Research Discovery Systems

The broader ecosystem of AI-assisted discovery systems highlights increasing interest in augmenting researchers with automated tools. Recent surveys Bolaños et al. [2024] show AI being applied to automate literature reviews, triage, and summarization, though human oversight remains essential. Recommender systems have applied link prediction to scholarly networks, showing that even simple graph features can forecast emerging research directions Krenn et al. [2023]. Systems like *Research-Link* and *HypoGen* Borrego et al. [2025], Qi et al. [2025] extend this to hypothesis generation, with promising results validated by expert judgment. Conversational assistants built on top of scholarly KGs Meloni et al. [2023] enable natural-language interaction grounded in citations and provenance, moving toward the notion of a "research concierge." Despite these advances, few systems provide an end-to-end pipeline that extracts open problems, organizes them as structured entities, links them relationally, and ranks them by novelty, tractability, or impact. This gap underscores the novelty of our approach, which explicitly targets the representation and prioritization of research problems as the core unit of discovery.

# 3 Research Questions and Hypotheses

The research is guided by the following questions and corresponding hypotheses:

- **RQ1:** What level of reliability can LLMs achieve in extracting research problem statements, assumptions, and constraints from heterogeneous academic papers, and how does this performance compare to human annotators? **HP1:** With structured prompting, schema validation, and lightweight human-in-the-loop review, LLMs will achieve extraction performance (precision, recall, F1) within 10% of human annotator agreement levels across a representative sample of papers.

- **RQ2:** In what ways does representing extracted problems as a graph with embeddings improve retrieval and linkage compared to text search alone, and what measurable gains can be observed in retrieval quality? **HP2:** A hybrid symbolic–semantic representation will significantly outperform citation- and keyword-based baselines, yielding higher mean reciprocal rank (MRR) and normalized discounted cumulative gain (nDCG) in retrieval tasks, as well as more accurate identification of extends/contradicts/depends-on relations.

- **RQ3:** Which ranking mechanisms (e.g., freshness, tractability, impact, community interest) most effectively surface "high-value" research opportunities, and how do researchers perceive their usefulness in practice? **HP3:** A ranking function that combines freshness decay,

tractability indicators (datasets, metrics, baselines), and impact proxies (cross-domain links, community interest) will produce results rated by users as more useful and trustworthy than baseline orderings, reducing task completion time and increasing satisfaction in user studies.

# 4 Methodology

## 4.1 Research Design

The study follows a systems design and evaluation approach, building a prototype and testing it against existing discovery workflows.

## 4.2 System Architecture

The proposed pipeline consists of:

1. Ingestion of papers (arXiv, OpenAlex).
2. Segmentation into sections (Intro, Methods, Results, Future Work).
3. Extraction of problem statements and constraints using LLMs with structured JSON schema.
4. Normalization and schema validation.
5. Storage in a property graph (Neo4j or Neptune) and vector index (pgvector/FAISS).
6. Retrieval and ranking using hybrid symbolic + semantic search.
7. Human-in-the-loop validation through a lightweight review UI.

A high-level view of the prototype architecture is provided in Figure 1 in Appendix A.

## 4.3 Data Collection

Initial focus will be on the domain of Software Engineering, using approximately 200 papers from major venues (e.g., ICSE, FSE, ASE, EMSE). Ground truth annotations will be collected for evaluation.

## 4.4 Data Analysis

Evaluation will include:

- **Precision/Recall:** of extracted problem statements against human annotations.
- **Linkage Quality:** correctness of extends/contradicts/depends_on relations.
- **User Study:** testing researcher satisfaction versus baseline tools (Google Scholar, Semantic Scholar).

# 5 Expected Outcomes

An overview of the proposed project and expected outcomes is provided in Appendix A, Table 1. The project will deliver:

1. A working prototype of an AI-driven research ideas graph, capable of ingesting papers, extracting structured problem statements, and supporting hybrid symbolic–semantic queries.

2. Benchmarks on extraction accuracy, relation quality, and retrieval effectiveness, directly addressing **RQ1/HP1** and **RQ2/HP2**:

   - *HP1 (Extraction reliability):* Quantitative evidence of what level of reliability LLMs can achieve in extracting problem statements, assumptions, and constraints, measured through precision, recall, and F1-score against human annotations. Error analysis will show how performance compares to human annotators across different paper types and sections.
   - *HP2 (Graph and embeddings for retrieval/linkage):* Empirical results showing in what ways a hybrid symbolic–semantic representation improves retrieval and linkage quality. Evaluation will include correctness of extends/contradicts/depends-on relations, mean reciprocal rank (MRR), and normalized discounted cumulative gain (nDCG) compared to citation- and keyword-based baselines.

3. Evidence of how AI can help organize and prioritize open research problems, corresponding to **RQ3/HP3**:

- **Quantitative evaluation:** Analysis of which ranking mechanisms (freshness, tractability, impact, community interest) most effectively surface high-value research opportunities. Improvements will be measured using nDCG, user-rated relevance scores, and coverage statistics.

- **User studies:** Structured studies with researchers (e.g., graduate students and faculty in Computer Systems) to assess how researchers perceive the usefulness of ranking mechanisms in practice. Metrics will include:
  - *Task completion time:* average time to identify relevant open problems with the prototype versus Google Scholar or Semantic Scholar.
  - *Perceived usefulness:* Likert-scale ratings (1–5) on clarity of problem representations, ease of navigation, and relevance of ranked results.
  - *Confidence in coverage:* percentage of participants who report discovering problems they would have otherwise overlooked.
  - *Satisfaction and trust:* Likert-scale ratings on whether provenance tracking and review features increase confidence in the extracted information.

- **Case studies:** Worked examples in the Computer Systems and Caching domain (e.g., cache replacement policies) demonstrating which ranking mechanisms bring hidden or underexplored problems to the surface, and how these insights can guide literature reviews and frame new experiments.

# 6 Limitations

While this proposal outlines a promising direction for AI-assisted research discovery, several limitations should be acknowledged.

## 6.1 Extraction Accuracy

The reliability of problem extraction depends heavily on the capabilities of large language models (LLMs). Despite advances, LLMs may misinterpret ambiguous phrasing, overlook implicit assumptions, or produce inconsistent structured outputs. Although schema validation and human-in-the-loop review help mitigate these risks, achieving high recall and precision across diverse academic writing styles remains challenging.

## 6.2 Domain and Corpus Coverage

The initial implementation will focus on a single domain (Computer Systems and Caching) and a limited corpus (approximately 200 papers). This scope is sufficient for proof-of-concept but restricts generalizability. Scaling to broader scientific domains will require addressing domain-specific vocabularies, heterogeneous publishing conventions, and increased computational demands.

## 6.3 Ranking and Evaluation

The proposed ranking mechanisms (e.g., freshness, tractability, impact, community interest) are proxies for true research value and may not capture all relevant dimensions. Empirical evaluation of ranking effectiveness will be limited to small-scale user studies, which may not reflect the diversity of researcher needs across fields.

## 6.4 Resource Constraints

The system relies on LLM inference, graph database hosting, and vector indexing, which require substantial compute resources. Budgetary and infrastructure limitations may constrain the size of the corpus ingested and the frequency of updates, potentially reducing the timeliness of surfaced opportunities.

## 6.5 Ethical and Provenance Challenges

Although the system emphasizes provenance through DOIs, quoted spans, and confidence scores, risks remain. Misrepresentations of author intent or errors in linking problem statements could propagate if not carefully curated. Furthermore, reliance on open-access corpora may bias the repository toward certain venues, limiting representativeness.

## 6.6 Agentic Extensions

Future extensions envision autonomous agents ranking problems and executing experiments. However, these capabilities raise questions about reproducibility, accountability, and mechanistic interpretability. At this stage, the proposal does not provide safeguards against unintended biases in how agents prioritize or interpret scientific problems.

Overall, these limitations highlight the need for cautious deployment, iterative evaluation, and ongoing collaboration with the research community to ensure the system's reliability, fairness, and scalability.

# 7 Future Work

This project establishes the foundation of a research ideas graph: a structured repository of problem statements, assumptions, constraints, datasets, and metrics Jaradeh et al. [2019], Dessì et al. [2020]. Future work will extend this foundation beyond passive storage toward an active ecosystem of autonomous agents that not only curate but also advance research.

## 7.1 Agent-Orchestrated Ranking and Prioritization

A first direction is the development of agents dedicated to ranking open problems stored in the repository. Ranking will integrate multiple dimensions, including:

- **Freshness and novelty:** recently proposed problems and shifts in citation patterns.
- **Tractability:** availability of datasets, clarity of evaluation metrics, and reproducibility.
- **Potential impact:** cross-domain connections and alignment with community interest.
- **Community signals:** citations, forks, bookmarks, and replication attempts.

These agents will surface "high-value" opportunities and generate watchlists or alerts when new evidence arises.

## 7.2 Autonomous Experimentation Agents

A second direction is the creation of agents capable of executing experiments derived from problem statements in the repository. Given a structured statement with datasets, metrics, and baselines, such agents can:

1. Generate candidate experimental designs.
2. Execute reproducible workflows (e.g., containerized environments or cloud pipelines).
3. Record results in a standardized schema for comparison and replication.

Prior work in autonomous experimentation demonstrates feasibility: robot scientists such as *Adam* and *Eve* have conducted closed-loop cycles of hypothesis generation, experimentation, and analysis in biology and drug discovery King et al. [2009, 2018]. More recent frameworks couple large language models with laboratory automation to design and execute experiments in chemistry and materials science Bran et al. [2024], Boiko et al. [2023]. These advances suggest that structured problem graphs can support autonomous computational experimentation in computer systems, program synthesis, and machine learning.

The extended architecture that incorporates autonomous agents for ranking, experimentation, and feedback is shown in Figure 2 in Appendix A.

## 7.3 Closed-Loop Research Cycles

Outputs from experiment-executing agents will not terminate in isolation. Instead, results will be:

1. Logged back into the repository as extended problem statements, refinements, or resolved conjectures.
2. Packaged into draft papers that follow scholarly conventions, ready for peer review.
3. Linked to future work signals, enabling subsequent agents to propose follow-up studies.

This design enables a self-sustaining research loop: repository → agents → experiments → new knowledge → repository.

## 7.4 Human-in-the-Loop Collaboration

While agents can automate ranking and experimentation, humans remain essential for oversight, creativity, and judgment. Future work will explore:

- Interfaces for researchers to guide agents by adjusting ranking criteria or suggesting baselines.
- Semi-automated peer review workflows, where agents generate structured reviews but humans validate.
- Continuous ingestion from digital libraries (e.g., OpenAlex, arXiv) to expand the problem graph with new literature.

## 7.5 Mechanistic Interpretability of Agent Decisions

A critical open question is not only whether agents can autonomously propose and execute experiments, but also *why* they select specific directions. Future work will apply mechanistic interpretability techniques Olah et al. [2020], Nanda et al. [2023] to probe the internal representations of ranking and experimentation agents. Specifically:

- **Circuit-level analysis:** Identify attention heads and pathways responsible for weighting problem features (e.g., dataset availability, citation freshness).
- **Decision decomposition:** Trace how embeddings of assumptions, constraints, and metrics influence final experiment selection.
- **Counterfactual probing:** Alter problem attributes (e.g., swap metrics or baselines) to measure causal influence on agent choice.
- **Transparency dashboards:** Expose interpretable explanations of agent reasoning to human collaborators, supporting trust and oversight.

This direction bridges autonomous research with explainable AI, ensuring that agent-driven experimentation is not a "black box" but a transparent process that researchers can interrogate, debug, and refine. Incorporating mechanistic interpretability aligns with broader goals of responsible AI and scientific accountability.

## 7.6 Toward Agentic Scientific Communities

Ultimately, the repository and its agents can serve as the nucleus of *agentic scientific communities*. Multiple specialized agents—problem finders, rankers, experimenters, reviewers—will interact in coordinated fashion, supervised by humans. This vision moves toward "AI-extended science," where humans focus on strategy and interpretation while delegating routine discovery and validation to autonomous agents.

# 8 Conclusion

This work introduces an AI-driven system for structuring and organizing open research problems into a navigable knowledge graph, enabling researchers to more efficiently identify gaps, compare contributions, and surface promising directions for inquiry. By combining symbolic representations with semantic retrieval, the system provides new ways of interacting with the rapidly growing body of scientific knowledge.

From a social perspective, the potential benefits are considerable. The platform democratizes access to research opportunities by reducing the entry barrier for students, early-career scholars, and researchers outside elite institutions. It can accelerate discovery by making open problems transparent and actionable, foster interdisciplinary collaboration by linking related challenges across domains,

and support funding agencies in identifying impactful areas for investment. More broadly, such a system may help sustain the pace of scientific innovation by transforming the overwhelming volume of publications into a coherent map of unresolved questions.

At the same time, the approach raises important concerns. Automating the identification and ranking of research problems risks encoding biases from the training data or from dominant publication venues, potentially reinforcing inequities in which problems are prioritized. Over-reliance on AI-generated suggestions could also narrow the diversity of inquiry, discouraging unconventional or speculative research directions. Furthermore, the collection and structuring of research outputs must respect copyright boundaries, researcher consent, and appropriate use of intellectual contributions.

Overall, while the proposed system has the potential to positively reshape how the research community engages with open problems, its deployment must be carefully guided by responsible AI practices. By balancing innovation with vigilance regarding social impacts, we aim to build a tool that not only accelerates discovery but also sustains the values of openness, inclusivity, and fairness in scientific research.

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

# A  Technical Appendices and Supplementary Material

**System Architecture Diagrams**

For clarity, we provide diagrams of the system in the prototype (Phase 1) and extended future work (Phase 2) configurations.

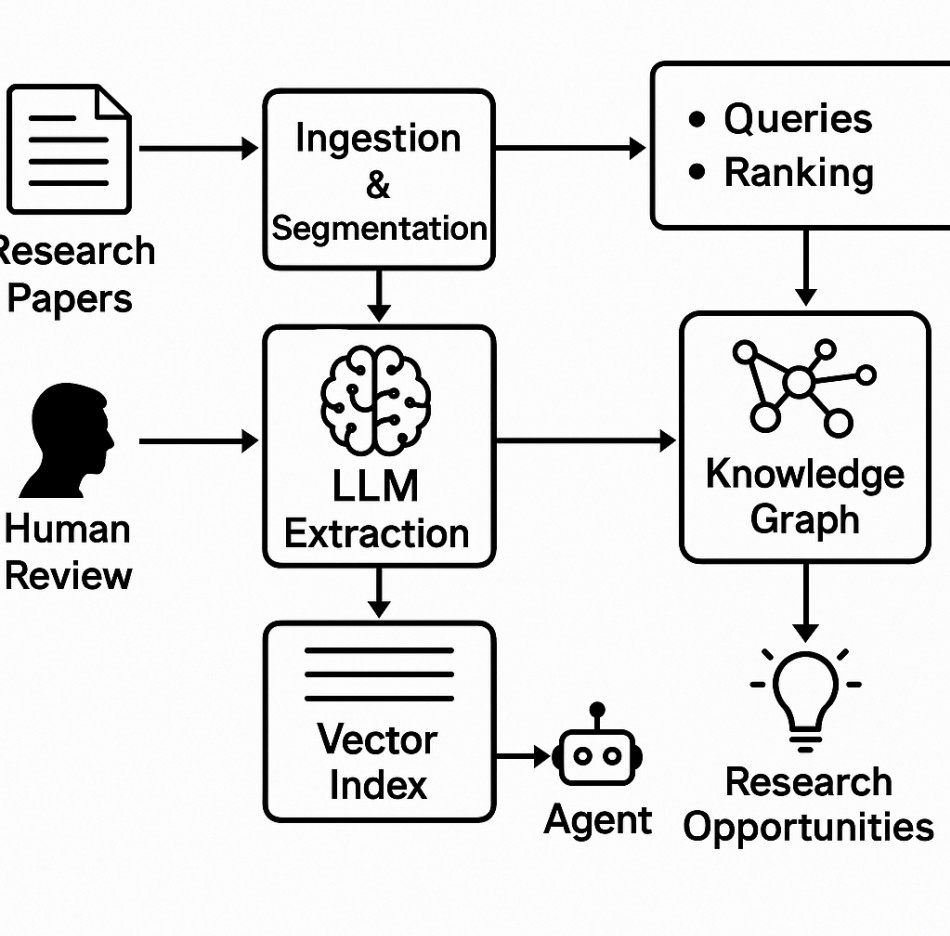

Figure 1: Phase 1: Prototype system architecture.

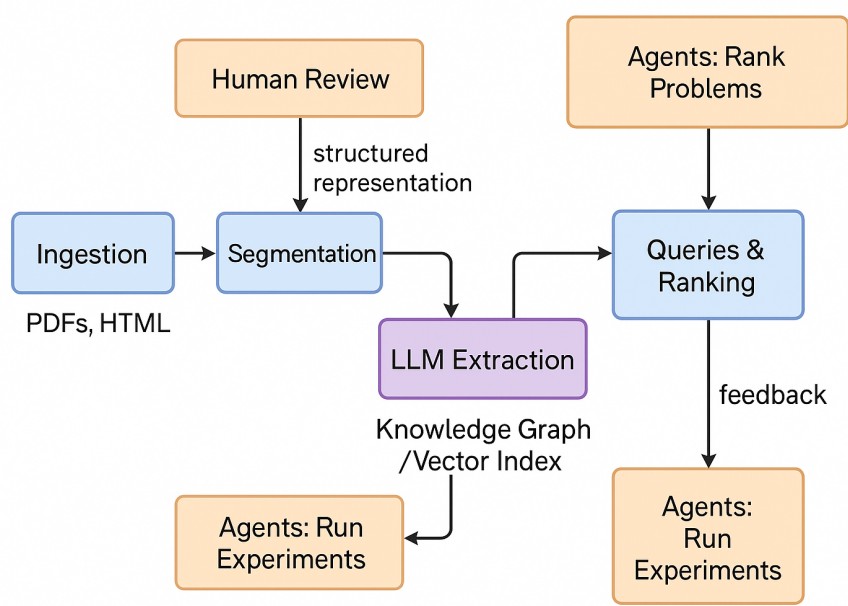

Figure 2: Phase 2: Extended architecture including ranking and experiment-executing agents.

Listing 1: Example structured representation of a research problem in software engineering.

```
402  {
403    "id": "prob:doi:10.1145/xxxx#p3",
404    "title": "Improving automated program repair under realistic bug distributions",
405    "problem_type": ["open_problem", "challenge"],
406    "statement": "Design an automated program repair technique that achieves higher co
407    "domain": ["Software Engineering", "Program Repair"],
408    "acm_ccs": ["D.2.5", "D.2.7"],
409    "assumptions": [
410      "Bugs are sampled from open-source repositories (e.g., Defects4J, Bugs.jar)",
411      "Patch validation uses regression test suites"
412    ],
413    "constraints": [
414      "Repair must complete within 1 hour of compute time",
415      "Generated patches must compile successfully"
416    ],
417    "datasets": [
418      {"name": "Defects4J", "id": "doi:10.1145/2591062.2591069"},
419      {"name": "Bugs.jar", "id": "doi:10.1109/MSR.2018.00-11"}
420    ],
421    "metrics": ["PatchCorrectness", "CompilationSuccess", "TimeToRepair"],
422    "baselines": ["GenProg", "PAR", "TBar"],
423    "signals": [
424      {"type": "gap", "text": "Current techniques struggle with multi-location bugs."}
425      {"type": "future_work", "text": "Explore hybrid approaches that combine search-b
426    ],
427    "evidence": [
428      {
429        "source": "doi:10.1145/xxxx",
430        "section": "Limitations",
431        "spans": ["L410-L431"]
432      }
433    ],
434    "links": {
```

```
435        "extends": ["prob:arXiv:2101.01234#p1"],
436        "contradicts": []
437    },
438    "confidence": 0.74,
439    "extracted_at": "2025-09-15",
440    "version": "ideas-graph@0.3.1"
441 }
```

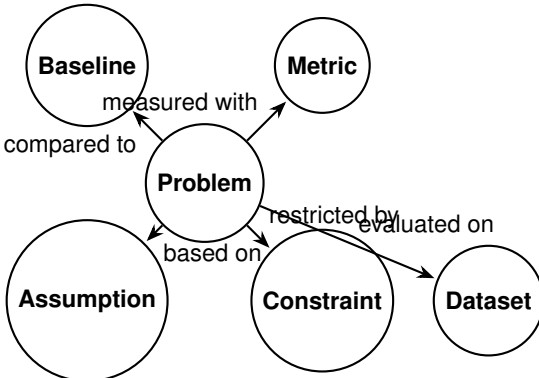

Figure 3: Illustrative knowledge graph structure: a problem connects to assumptions, constraints, datasets, metrics, and baselines.

Table 1: Mapping of Research Questions, Hypotheses, Expected Outcomes, and Metrics

| Research Question | Hypothesis | Expected Outcome | Metrics / Evidence |
|---|---|---|---|
| **RQ1:** What level of reliability can LLMs achieve in extracting research problem statements, assumptions, and constraints, and how does this performance compare to human annotators? | **HP1:** With structured prompting, schema validation, and human-in-the-loop review, LLMs will achieve extraction performance within 10% of human agreement levels. | Benchmarks showing extraction accuracy and reliability of LLMs relative to human annotators. | Precision, recall, F1-score; error analysis by paper type and section. |
| **RQ2:** In what ways does representing extracted problems as a graph with embeddings improve retrieval and linkage compared to text search alone, and what measurable gains can be observed? | **HP2:** A hybrid symbolic–semantic representation will outperform citation- and keyword-based baselines in retrieval quality and relation accuracy. | Demonstrated improvements in retrieval effectiveness and correctness of extends/contradicts/depends on relations. | Mean Reciprocal Rank (MRR), normalized Discounted Cumulative Gain (nDCG), relation correctness rate. |
| **RQ3:** Which ranking mechanisms (freshness, tractability, impact, community interest) most effectively surface high-value research opportunities, and how do researchers perceive their usefulness in practice? | **HP3:** A combined ranking function will surface problems judged more useful and trustworthy by users, reducing task time and improving satisfaction. | Evidence from quantitative ranking evaluation, user studies, and domain case studies. | nDCG, user-rated relevance, task completion time, Likert-scale ratings (usefulness, trust), coverage confidence, case study demonstrations. |

## B   Ethical Considerations

The system will ensure provenance by storing DOIs and quoted spans, include confidence scores to mitigate misrepresentation, and use open-access corpora to respect copyright. All extracted problem statements, assumptions, and constraints are linked to their original sources to maintain transparency and prevent misuse

**Responsible AI and Broader Impact Statement**

This work complies with the NeurIPS Code of Ethics. The proposed system seeks to democratize access to scientific knowledge by lowering barriers to identifying and extending open research problems. The broader impact includes enabling students, early-career researchers, and under-resourced institutions to navigate research more effectively. Potential risks include bias in ranking research opportunities or misrepresentation of extracted content. To mitigate these, we (1) restrict our corpus to open-access publications, (2) provide full provenance (DOIs, spans, confidence scores), and (3) integrate human-in-the-loop review to safeguard against errors. Our intent is to augment human reasoning rather than replace it, ensuring responsible deployment of the AI scientist.

**Reproducibility Statement**

We have made explicit efforts to support reproducibility. The extraction pipeline is defined through a strict JSON schema with validation rules, ensuring deterministic outputs. We will release code, prompts, schema definitions, evaluation datasets, and annotation guidelines under an open-source license. All experiments will be documented with configuration files, hyperparameters, and software versions to allow faithful replication. Evaluation metrics (precision/recall, linkage quality, user study protocols) will be fully described, enabling independent verification and extension of our results.

# C    Agents4Science AI Involvement Checklist

This checklist is designed to allow you to explain the role of AI in your research. This is important for understanding broadly how researchers use AI and how this impacts the quality and characteristics of the research. **Do not remove the checklist! Papers not including the checklist will be desk rejected.** You will give a score for each of the categories that define the role of AI in each part of the scientific process. The scores are as follows:

- **[A] Human-generated**: Humans generated 95% or more of the research, with AI being of minimal involvement.

- **[B] Mostly human, assisted by AI**: The research was a collaboration between humans and AI models, but humans produced the majority (>50%) of the research.

- **[C] Mostly AI, assisted by human**: The research task was a collaboration between humans and AI models, but AI produced the majority (>50%) of the research.

- **[D] AI-generated**: AI performed over 95% of the research. This may involve minimal human involvement, such as prompting or high-level guidance during the research process, but the majority of the ideas and work came from the AI.

1. **Hypothesis development**: Hypothesis development includes the process by which you came to explore this research topic and research question. This can involve the background research performed by either researchers or by AI. This can also involve whether the idea was proposed by researchers or by AI.

    Answer: **[B]**

    Explanation: I came up with the idea and the process. ChatGPT modified parts of it based on it's own deep research. It decided that a graph data structure was a better option than the original JSON files that I had suggested.

2. **Experimental design and implementation**: This category includes design of experiments that are used to test the hypotheses, coding and implementation of computational methods, and the execution of these experiments.

    Answer: **[B]**

    Explanation: Since this is just a proposal for a 9 month idea, we currently have not executed experiments, however, this proposal is about a process that allows agents to continuously experiment and further research.

3. **Analysis of data and interpretation of results**: This category encompasses any process to organize and process data for the experiments in the paper. It also includes interpretations of the results of the study.

    Answer: **[B]**

    Explanation: We did not execute experiments in this paper as it is a proposal. However, as part of this proposal we will have the AI agents analyze the data and results for intpretation and feedback by a human. This would be the human in the loop piece.

4. **Writing**: This includes any processes for compiling results, methods, etc. into the final paper form. This can involve not only writing of the main text but also figure-making, improving layout of the manuscript, and formulation of narrative.

    Answer: **[B]**

    Explanation: All writing in this paper was done by AI via a back and forth discussion with a human. Results pasted in to the submission.

5. **Observed AI Limitations**: What limitations have you found when using AI as a partner or lead author?

    Description: The main limitation I have run into so far is it's ability to go into detail on its own without further prompting. Part of my proposal is to build the agents in such a way that they will be able to act on their own for the most part with human interactions for approvals, and direction guidance, and less on the hand holding.

