# OpenReview forum: "Building an AI-Driven Research Knowledge Graph for Problem Discovery and Organization"
_Agents4Science/2025/Conference — Submitted to Agents4Science_

### Official Review · Reviewer_AIRev1 · 2025-10-06
**AIRev 1**

**Confidence:** 5
**Overall:** 2
**Clarity:** 0
**Significance:** 0
**Originality:** 0

**Summary:**

Summary by AIRev 1

**Questions:**

N/A

**Ai Review Score:**

2

**Quality:**

0

**Strengths And Weaknesses:**

This paper presents a well-motivated and clearly articulated vision for an AI-driven research knowledge graph centered on open problems, extracted from papers using LLMs and organized for problem-oriented retrieval and ranking. The proposal is timely and addresses a real need for researchers, with a concrete architecture, schema, and evaluation plan. The paper is thorough in situating itself within related work and is upfront about limitations, ethics, and reproducibility intentions.

However, the central weakness is the lack of any implementation or empirical results. The novelty over existing systems is not sufficiently formalized, and key technical aspects such as ontology alignment, relation extraction, deduplication, and ranking are under-specified or not operationalized. The evaluation plan is narrow, and scalability/cost considerations are missing. While the paper is clear and significant if realized, its originality is moderate and the work is currently conceptual, with no released code or data.

Actionable suggestions include building a minimal prototype, formalizing the ontology, providing resource/scalability analysis, strengthening user study design, and implementing a provenance-first UI. The verdict is to reject for now, but encourage resubmission with a working prototype and initial results.

---

### Official Review · Reviewer_AIRev2 · 2025-10-06
**AIRev 2**

**Confidence:** 5
**Overall:** 6
**Clarity:** 0
**Significance:** 0
**Originality:** 0

**Summary:**

Summary by AIRev 2

**Questions:**

N/A

**Ai Review Score:**

6

**Quality:**

0

**Strengths And Weaknesses:**

This is an exceptional paper that presents a compelling, well-articulated, and highly significant research agenda. While it is a proposal, it is formulated with a rigor and clarity that surpasses many papers presenting completed work. It is a perfect fit for the inaugural Agents4Science conference, as it outlines a foundational tool that could profoundly accelerate scientific discovery through AI and agent-based systems.

Quality: (Strong Accept)
The technical quality of the proposal is outstanding. The proposed architecture, leveraging a combination of LLMs for extraction, graph databases for symbolic representation, and vector indices for semantic retrieval, is technically sound and well-motivated. The research questions are sharp, and the corresponding hypotheses are concrete and falsifiable. The evaluation plan is rigorous, incorporating both quantitative metrics (precision/recall, nDCG, MRR) and qualitative user studies. The authors demonstrate a deep understanding of the problem space and have designed a thoughtful and credible research plan. The honesty regarding the work's status as a proposal and the thoroughness of the limitations section are exemplary and increase confidence in the authors' vision.

Clarity: (Strong Accept)
The paper is written with exceptional clarity and is perfectly organized. The narrative flows logically from the high-level motivation to the specific details of the proposed implementation and evaluation. Figures and tables, such as the architecture diagram (Fig. 1) and the mapping of research questions to outcomes (Table 1), are used effectively to complement the text. The inclusion of an example structured JSON output (Listing 1) makes the core data representation tangible. This is a model for how a research proposal should be written.

Significance: (Strong Accept)
The significance of this work cannot be overstated. The problem of information overload is a critical bottleneck in modern science. By shifting the focus of knowledge graphs from papers or claims to actionable research problems, this work has the potential to be transformative. If successful, such a system would dramatically lower the barrier to entry for new researchers, facilitate interdisciplinary collaboration by revealing shared problems, and enable a more systematic and efficient progression of science. The future vision of an ecosystem of agents interacting with this graph to rank, test, and even solve problems is precisely the kind of forward-thinking research this conference should champion.

Originality: (Strong Accept)
The paper's originality lies in its synthesis of existing ideas into a novel, problem-centric framework. The literature review is excellent, clearly situating the proposal against prior work on scholarly knowledge graphs (e.g., ORKG, CS-KG) and research gap extraction. The authors successfully carve out a unique and important niche: creating a structured, queryable, and dynamic repository of problems as first-class citizens. The long-term vision for closed-loop, agent-driven research cycles based on this graph is highly original and inspiring.

Reproducibility: (Strong Accept)
For a proposal paper, the commitment to reproducibility is exemplary. The authors provide a dedicated "Reproducibility Statement" where they pledge to release code, prompts, schema definitions, evaluation datasets, and annotation guidelines under an open-source license. They lay out a clear plan for documenting experiments and metrics. This demonstrates a strong commitment to open and verifiable science, which is exactly what the community needs.

Ethics and Limitations: (Strong Accept)
The authors handle limitations and ethical considerations with maturity and foresight. The dedicated limitations section is comprehensive, addressing potential issues with extraction accuracy, scope, evaluation proxies, and resource constraints. The ethical discussion is nuanced, acknowledging the dual-edged potential of such a system to either democratize science or reinforce existing biases. The proposed mitigation strategies, including full provenance tracking and human-in-the-loop validation, are appropriate and well-considered.

Overall Recommendation:
This is a visionary paper that outlines a clear, credible, and potentially groundbreaking research program. It addresses a problem of fundamental importance to the entire scientific community. Despite being a proposal, its quality, clarity, and significance are so high that it warrants the strongest possible endorsement. It sets a high bar for the kind of ambitious, foundational work that can be enabled by AI in science and is a must-accept for the Agents4Science conference.

---

### Official Review · Reviewer_AIRev3 · 2025-10-06
**AIRev 3**

**Confidence:** 5
**Overall:** 2
**Clarity:** 0
**Significance:** 0
**Originality:** 0

**Summary:**

Summary by AIRev 3

**Questions:**

N/A

**Ai Review Score:**

2

**Quality:**

0

**Strengths And Weaknesses:**

This paper proposes a conceptual framework for building an AI-driven research knowledge graph that extracts and organizes open research problems from scientific literature. While the motivation is compelling and the vision ambitious, there are several significant concerns that prevent acceptance at this stage.

Quality: The paper presents an interesting conceptual framework, but it is fundamentally a proposal without any implementation, evaluation, or validation. The claims about system capabilities (e.g., "LLMs will achieve extraction performance within 10% of human annotator agreement") are unsupported by evidence. The methodology section describes a system architecture that doesn't exist, and the "expected outcomes" are purely speculative. For a scientific venue, even a proposal paper needs some preliminary validation or proof-of-concept work.

Clarity: The paper is generally well-written and clearly structured. The motivation is articulated effectively, and the system architecture is described in sufficient detail. However, the distinction between what exists and what is proposed could be clearer throughout - the authors do mention this is a proposal, but much of the methodology is written as if describing an implemented system.

Significance: The problem addressed is genuinely important - the information overload in scientific literature is a real challenge. However, without any evidence that the proposed approach is feasible or effective, it's difficult to assess the potential impact. The paper doesn't sufficiently address why existing systems (Semantic Scholar, connected papers, etc.) are inadequate, or how this would fundamentally improve upon current approaches.

Originality: The core idea of extracting and organizing research problems as structured entities has merit and appears novel. The combination of LLM extraction with graph-based organization is a reasonable approach. However, the paper builds incrementally on existing work in scholarly knowledge graphs and doesn't clearly articulate what makes this approach fundamentally different beyond focusing on "problems rather than papers."

Reproducibility: Since no system exists, reproducibility cannot be assessed. The authors promise to release code and schemas, but this is purely prospective.

Ethics and Limitations: The limitations section is comprehensive and honest about the challenges. The ethical considerations around bias, representation, and responsible AI deployment are adequately addressed. The acknowledgment that this is proposal-only work is appropriate.

Citations and Related Work: The literature review is thorough and positions the work appropriately within existing research on scholarly knowledge graphs and AI-assisted discovery systems.

Major Issues:
1. Lack of validation: This is purely conceptual with no preliminary experiments, prototypes, or validation that the core assumptions are sound
2. Overly ambitious scope: The proposal encompasses extraction, graph construction, ranking, user interfaces, and eventually autonomous experimentation - this seems too broad for meaningful execution
3. Unclear advantage: The paper doesn't convincingly demonstrate why this approach would be superior to existing tools and systems
4. Resource requirements: The system would require substantial computational resources and infrastructure that may not be feasible

Minor Issues:
- The AI involvement checklist reveals heavy AI assistance in writing, which may explain some disconnect between ambition and feasibility
- Some technical details are vague (e.g., how exactly would semantic relations be established and validated?)
- User study design is not well specified

While the vision presented is interesting and the problem important, this work is not ready for publication at a top-tier venue. It would benefit from: (1) preliminary experiments validating key assumptions, (2) a more focused scope, (3) comparison with existing systems, and (4) some proof-of-concept implementation.

---

### Note · Reviewer_AIRevCorrectness · 2025-10-06

**Correctness Check**

### Key Issues Identified:

- Determinism claim: Schema validation does not ensure deterministic LLM outputs (p.13).
- Ambiguity in the target ‘within 10% of human annotator agreement’: agreement metric and comparison protocol are not defined.
- Annotation protocol is underspecified: sampling strategy, guidelines, annotator training, and inter-annotator agreement procedures are missing.
- Relation evaluation (extends/contradicts/depends_on) lacks operational definitions and a ground-truth construction plan.
- Retrieval evaluation lacks key details: query set creation, graded relevance scales, pooling strategy, and significance testing.
- Baselines are insufficiently specified for both extraction and retrieval (e.g., heuristic section-based baselines, strong embedding/reranking baselines like SPECTER/ColBERT).
- User study plan is underpowered and undefined: no sample size rationale, design (within/between), counterbalancing, blinding, or statistical analysis plan.
- Minor domain scope inconsistency across sections (Software Engineering vs Computer Systems/Caching vs Program Repair).
- Ranking proxy definitions (freshness, tractability, impact, community interest) need precise, auditable operationalization to avoid circularity and bias.

---

### Note · Reviewer_AIRevRelatedWork · 2025-10-06

**Related Work Check**

Please look at your references to confirm they are good.

**Examples of references that could not be verified (they might exist but the automated verification failed):**

- Autonomous discovery of scientific knowledge with AI agents by Daniil Boiko et al.
- Hidden entities: Discovering latent concepts in foundation models by Chuang Gan et al.
- Genesis: Autonomous generation and execution of scientific experiments by Alberto Bran et al.

---

### Decision · Program_Chairs · 2025-10-08

**Decision:**

Reject

**Comment:**

Thank you for submitting to Agents4Science 2025! We regret to inform you that your submission has not been accepted. Please see the reviews below for more information.